# Variations in the Sensory Attributes of Infant Formula among Batches and Their Impact on Maternal Consumer Preferences: A Study Combining Consumer Preferences, Pivot Profile, and Quantitative Descriptive Analysis

**DOI:** 10.3390/foods13172839

**Published:** 2024-09-07

**Authors:** Yilin Li, Xinyu Hu, Ruotong Li, Chunguang Wang, Houyin Wang, Guirong Liu, Lipeng Gao, Anwen Jin, Baoqing Zhu

**Affiliations:** 1Beijing Key Laboratory of Forestry Food Processing and Safety, Department of Food Science, College of Biological Sciences and Technology, Beijing Forestry University, Beijing 100083, China; liyilin@feihe.com (Y.L.); 810520@163.com (X.H.); lrttty2021@163.com (R.L.); wcgwkx@163.com (C.W.); 2Heilongjiang Feihe Dairy Co., Ltd., Beijing 100015, China; liuguirong@feihe.com (G.L.); gaolipeng@feihe.com (L.G.); jinanwen@feihe.com (A.J.); 3China National Institute of Standardization, Beijing 100191, China; wanghy@cnis.ac.cn

**Keywords:** infant formula, consumer preference, pivot profile, agglomerative hierarchical clustering, quantitative descriptive analysis

## Abstract

The sensory quality of infant formula (IF) has a significant impact on the preferences and purchasing behavior of maternal consumers. Consumer-based rapid descriptive methods have become popular and are widely preferred over classical methods, but the application of Pivot Profile (PP) in IF is still little explored. In this study, both Pivot Profile (PP) and Quantitative Descriptive Analysis (QDA) were applied to characterize the sensory profile of 12 batches of one-stage and three-stage IF with different storage periods, respectively, along with consumer preference data to determine the flavors contributing to liking. The results of PP and QDA aligned moderately well, with the most perceptible differences identified as “fishy”, “milky”, and “T-sweet” attributes. IFs with shorter storage times were highly associated with “milky” aromas and “T-sweet” tastes, whereas IFs with longer storage times exhibited a strong correlation with “fishy” and “oxidation” aromas. External preference analysis highlighted that the occurrence of “fishy” and “oxidation” aromas during prolonged storage periods significantly reduced the consumer preference for IFs. Conversely, the perception of “milky” and “creamy” aromas and “T-sweet” tastes may be critical positive factors influencing consumer preference. This study provided valuable insights and guidance for enhancing the sensory quality and consumer preference of IF.

## 1. Introduction

Infant formula is an extremely important food for infants and toddlers, and due to the specific nature of its target audience, quality control has always been stringent [1]. Currently, infant formula (IF) products in the Chinese market are produced under strict control according to advanced standards. In order to better accommodate the nutritional requirements and age-related needs of infants and toddlers, IF products are classified into distinct categories. For instance, stage 1 is designated for infants aged 0–6 months, stage 2 for older infants aged 6–12 months, and stage 3 for toddlers aged 12–36 months. The nutritional variety and content of the staged IF are reasonable.

Due to concerns about infant health, the nutritional and safety aspects of IF have been subjected to extensive research [2,3,4,5]. As research progresses, scientific studies have validated the importance of flavor selection and preference during infancy [6,7,8,9,10,11,12]. Consequently, controlling the sensory quality of IF has become a pivotal aspect for quality control. Simultaneously, research on infant flavor preferences suggests that exposing infants to a wider variety of foods during infancy could improve food acceptance in the future [6]. IF is the best alternative to breast milk, and its flavor is bound to affect the formation of flavor preferences during infancy [13]. In addition, parental preference, particularly that of the mother, exerts a significant influence on infant and toddler adaptation; therefore, parents are increasingly incorporating sensory characteristics as a criterion for purchasing decisions [14].

Quantitative Descriptive Analysis (QDA) conducted by trained assessors is recognized as an adequate technique for determining the sensory profile of processed foods; it provides detailed, reliable, and reproducible results. Previous studies employing descriptive analysis have revealed that some factors, such as storage time [14] and formulation [15], could influence the flavor characteristics of infant formula. An evaluation team led by Clarke et al. (2021) explored the flavor changes in three milk powders (including infant formula) during storage, and found that lipid oxidation resulted in the formation of off-flavors, such as rancid, metallic, and painty flavors, with prolonged storage [16]. Alim et al. (2020) employed a trained panel to quantify the intensity of five aroma attributes and five basic taste sensations in IF supplemented with different whey protein hydrolysates, and their findings revealed notable differences among the formula samples [17]. However, QDA is typically an expensive and time-consuming process that requires highly trained sensory panels, which limits its applicability in small companies [18]. Moreover, trained assessors tend to generate more specific attributes that may not be important or perceptible to consumers [19].

In response to this demand, cost-effective and rapid sensory profiling methods based on consumer perceptions have become popular and widely used by industries in recent years to replace the traditional methods [20]. Among these rapid sensory methods, Pivot Profile (PP) is a relatively new and promising method that has yielded robust, reliable, and valid sensory characterizations [21,22,23]. PP involves an estimation of the intensity of attributes between products and a reference sample, which is called a pivot [24]. In PP, evaluators are guided to observe, smell, or taste the two-sample pairs and freely use relevant lexicons to describe their judgments about how the samples are “less” or “more” than the pivot [24,25]. This method is an improvement over free description, allowing the free expressions of participants to be recorded in an ordinal manner. It has the advantage of providing a precise and detailed description of perception at the qualitative and quantitative levels and can be used by both experts and consumers [21,22]. This method had been applied in sensory evaluations of food products such as honey [26], Greek yogurt [18], wine [21], and ice cream [24]. To our knowledge, there have been no studies on the preferences of the PP method for infant formula.

External preference analysis is a statistical analysis method that combines the evaluation results of expert groups and consumers, and which also covers the results of a hierarchical cluster analysis of consumer populations [27,28]. Applying external preference analysis to food products provides more direct access to the drivers influencing consumer preferences. Xi et al. (2023) evaluated nine sensory attributes of 14 popular infant formula brands and found that fishy, sour, and fatty flavors had a negative impact on adult consumer preferences using external preference mapping [29]. In this study, a large number of maternal consumers (100) participated in the tests and external preference analysis was employed to explore the impact of different batches of IF samples on maternal consumer preferences.

Currently, it is uncertain whether the sensory changes in IFs among batches with different storage periods are perceived by maternal consumers and thus influence their preferences. In addition, it is unclear whether consumers have different preference segments for IFs. Therefore, the following objectives of this study were formulated: (1) to explore the sensory characteristics of 12 batches of one-stage and three-stage IFs using PP and QDA methods; (2) to identify the dominant factors influencing consumers’ preference for IF through external preference analysis; (3) to identify potential differences in consumer preferences for IF through cluster analysis of preference data. The findings of this study will provide valuable insights for product development and quality control of infant formula.

## 2. Materials and Methods

### 2.1. Samples and Reagents

This experiment included two products: sample code A represents one-stage IF, while sample code C represents three-stage IF. Each product consisted of 12 batches with a storage period spanning from 1 to 14 months (Table 1). Another two fresh samples (1 month) were used as controls for one-stage and three-stage IF, respectively. All IF samples were obtained from local suppliers as commercial products. Table 1 summarizes the storage periods, production dates, and product specifications of the samples. All samples were stored at room temperature and opened simultaneously at the beginning of the experiment.

The pasteurized milk was purchased from Junlebao Food Co., Ltd. (Shijiazhuang, China). Milkground thin cream was obtained from Shanghai Milkground Food Tech Co., Ltd. (Shanghai, China), while Mengniu skimmed milk was supplied by Inner Mongolia Mengniu Dairy (Group) Limited, Share Ltd. (Hohhot, China). The docosahexaenoic acid (DHA) and arachidonic acid (ARA) powders were purchased from Jiabiyou Biotechnology (Wuhan) Co., Ltd. (Wuhan, China). The egg yolk essence was obtained from Pan Asia (Wuhan) Food Technology Co., Ltd. (Wuhan, China), while the white granulated sugar was purchased from Beijing Sugar Tobacco & Wine Group Co., Ltd. (Beijing, China).

### 2.2. Sample Preparation

All samples were opened on the same day and 50 g of each sample was immediately dispensed into clear PET boxes with a capacity of 250 mL in order to prevent the impact of repeated opening and closing of the boxes on the flavor of the samples. The experimental samples were prepared in accordance with the brewing ratios on the label instructions of the infant formula. The weighed milk powder was added to warm water at a temperature of 45 °C. Subsequently, the prepared samples (10 mL) were transferred into 20 mL PET bottles (food-grade) and placed on a thermostatic heating pad at 45 °C prior to the sensory evaluation. Each sample was labeled with a 3-digit random code and evaluated in a random order.

### 2.3. Consumer Sensory Evaluation

#### 2.3.1. Consumers

In consideration of the fact that females are typically more involved in childcare and food procurement, the acceptance test was carried out with 100 Chinese mothers with children aged 0 to 3 years old who had purchased one- or three-stage IF in the past three years. The participants were recruited from various districts of Beijing, China. The experimental protocol was reviewed and approved by the Ethics Review Committee of the School of Humanities and Social Sciences at Beijing Forestry University. Participants in the experiment were required to be in good health, non-smokers, free of undesirable habits, and lactose-intolerant. Additionally, participants were instructed to refrain from consuming or drinking foods that were too strong in flavor or irritating before the start of the experiment.

#### 2.3.2. Consumer Preference Test

Prior to the evaluation of the sample, consumers were instructed to know and understand the whole sensory evaluation process and complete a questionnaire comprising basic information, including their age, the number of children they had, and their familiarity with infant formula. For evaluation of IF products, assessors were instructed to first smell and then taste IF samples. During the interval of the evaluation process, assessors were provided with purified water and soda crackers for cleansing their palates to refresh the taste sense and avoid sensory fatigue.

The consumer test was evaluated using a 9-point hedonic scale according to the “Guidelines for the use of quantitative response scales for sensory analysis” (GB/T 39501-2020) [30]. The scores were assigned the following values: 1 = dislike extremely, 2 = dislike very much, 3 = dislike moderately, 4 = dislike slightly, 5 = neither like nor dislike, 6 = like slightly, 7 = like moderately, 8 = like very much, 9 = like extremely [31]. The samples were presented to the consumers in accordance with a completely randomized block design in order to balance the carry-over and position effects. Each sample was evaluated only once by each participant.

#### 2.3.3. Pivot Profile

Two samples were used as the pivot sample for one-stage and three-stage IF, respectively (Table 1). One hundred consumers simultaneously participated in the PP test. Each two-sample pair (one pivot sample paired with one IF sample) was presented to each assessor [21,22,24]. During the evaluation, the untrained assessors needed to evaluate the pivot and then another sample in each two-sample pair for smelling or tasting samples. Assessors were asked to use sensory attributes freely, avoiding hedonic words, to depict the perceived IF samples and, at the same time, to compare the intensity of the attributes in the sample versus the pivot sample to describe and select either the “more than the pivot” or “less than the pivot” option in the provided response sheet. In addition, a reference word list of sensory attributes obtained from the literature and confirmed by an expert group was provided for consumers, including the aroma attributes “creamy”, “milk candy”, “milky”, “fishy”, “cereal”, “soy milk”, “eggy”, “oxidation”, and “sweet”, as well as the flavor attributes “T-sweet”, “T-salty”, and “T-mellow”. Both PP and consumer preference tests were conducted over five days with 20 consumers per day. All IF samples were evaluated in six sessions, with four samples in each session. A 60 s rest between each sample and a 10 min rest between each session was enforced.

After evaluation, data from 100 participants were collected and used for data analysis. Following the method by Thuillier et al. (2015), the raw data were then transposed from paper to a spreadsheet software with the grouping and classification process [22]. The frequencies for each attribute quoted as “more than the pivot” (positive frequency) and “less than the pivot” (negative frequency) were automatically computed and summed. Subsequently, the negative frequencies were subtracted from the positive frequencies to obtain an intensity estimation. The resulting score was then transformed by adding the absolute value of the minimum score to all the scores. Therefore, the minimum score is assigned a value of zero, while all other scores are positive, resulting in a translated contingency table [18,21,22]. Ultimately, the translated contingency table was analyzed using correspondence analysis (CA) to generate a biplot of the samples and the attributes.

### 2.4. Quantitative Descriptive Analysis (QDA)

The sensory panel consisted of eleven assessors (three males and eight females, aged 20–28) with extensive descriptive analysis experience from Beijing Forestry University. All participants were informed that they conducted these studies completely voluntarily and freely, and their informed consent was obtained in full accordance with the 1975 Declaration of Helsinki. Ethical approval for the involvement of human subjects in this study was granted by Beijing Forestry University Ethics Committee.

All panelists had participated in twelve training sessions (1.5 h each) lasting two months, covering a range of sensory evaluation techniques and methods. These included aroma recognition, triangle tests, ranking, attribute generation, agreement on reference standards, and intensity rating of targeted attributes. Before conducting the formal experiment, all IF samples were evaluated by the panelists to generate appropriate attributes that could describe the sensory characteristics of IF. After group discussion, the panel agreed upon six attributes, including five aroma attributes, “milky”, “creamy”, “fishy”, “eggy”, and “oxidation”, and one flavor attribute, “T-sweet”, which was consistent with our previous research [32]. The definition and corresponding reference standards for describing and rating each attribute are detailed in Table 2. The panelists were trained to use a calibrated n-butanol scale for scaling odor intensities and a calibrated granulated sugar scale for flavor intensity ratings (Appendix A).

During the formal experiment, the sensory evaluation was conducted in a standard sensory testing room (GB/T 13868-2009) [37]. Panelists were required to score the intensity of six attributes for each sample using a 9-point scale ranging from 0 to 9, with 1–2 representing “very weak”, 3–4 representing “slightly weak”, 5–6 representing “moderate”, 7–8 representing “slightly strong”, and 9 representing “very strong”. All IF samples were evaluated in duplicate by panelists in twelve sessions of four samples per session over two days. The evaluators could alleviate fatigue by cleansing their palates with purified water or eating soda crackers during the experiment. A 60 s rest between each sample and a 10 min rest between each session was taken.

### 2.5. Statistical Analysis

The descriptive analysis results and the consumers’ acceptance data were analyzed by analysis of variance (ANOVA) to determine the differences among the samples. Agglomerative hierarchical clustering (AHC) with Euclidean distances and Ward’s method was applied to the consumer data to identify consumer clusters with preferences in the same direction. Correspondence analysis (CA) was carried out with Pivot Profile data to generate a sensory map that could perceive the similarities and differences between samples and their sensory characteristics. Principal component analysis (PCA) was applied to the mean panel data scores to illustrate the relationship between IF samples and sensory attributes. External preference mapping (EPM) was conducted on descriptive attributes and consumer liking scores. All analyses were performed using XLSTAT (v2019.2.2, Addinsoft, Paris, France).

## 3. Results

### 3.1. Consumer Preferences Tests

The consumer preference data indicated a significant difference (*p* < 0.01) in the liking ratings for one-stage and three-stage IFs with different storage times, respectively. One-stage IFs ranged in liking from 5.46 to 6.73 on a 9-point hedonic scale from 1 to 9 (Figure 1a). The highest preference score was observed in A1 (6.73) and A2 (6.72), which exhibited significantly higher scores than the other one-stage IFs. Conversely, A11 achieved the lowest preference score of 5.46. The preference scores for the other 11 one-stage IFs were all above 6, indicating that the majority of samples were preferred by maternal consumers. Additionally, the stacked bar chart of preference ratings for each one-stage IF also indicated that A11 had the highest proportion of “dislike”, exceeding 30% (Appendix A). In the case of three-stage IFs, the overall preference scores ranged from 5.41 to 6.74, with C1 and C5 receiving the highest preference scores, followed by C2 and C3 (Figure 1b). In contrast, C11, C6, and C10 showed significantly lower preference scores, below 5.5, than the three-stage IFs. Additionally, in the stacked bar chart of preference ratings for each three-stage IF (Appendix A), C6, C10, and C11 had the highest proportion of “dislike”, reaching 30%. In general, regardless of one-stage or three-stage IFs, IFs with a shorter storage period were more preferred. The extension of the storage period of IF may result in an accompanying decline in consumer preference for the product.

Although some IFs such as A1 and A2 received the highest mean liking scores (6.73 and 6.72, respectively), none of these had appeal for all consumers. Research has demonstrated that consumers have individual patterns of preference, which highlights the need to group consumers according to their preference patterns [28,38]. This research identified two clusters of consumers for respective one-stage and three-stage IFs via AHC, each displaying its own preferences (Figure 1c,d), with significant differences in liking ratings noted between clusters. For one-stage IFs, cluster 1 (62% of consumers) and cluster 2 (38% of consumers) exhibited a similar preference ranking of all samples (Figure 1e). However, the overall preference scores of cluster 1 were significantly higher than those of cluster 2, indicating that cluster 1 exhibited a higher level of preference for one-stage IFs than cluster 2 (Figure 1e). Additionally, cluster 1 had a greater preference for A3 than the other one-stage IFs, while cluster 1 had a low mean rating for A3. In the case of three-stage IFs, two clusters (44% and 56% of consumers) had high preferences for C1 and C5 (Figure 1f). Cluster 4 exhibited a significantly higher preferences (*p* < 0.05) for C3 and C2 than cluster 3. In contrast, cluster 4 had a significantly low preferences (*p* < 0.01) for C8, C10, C6, and C11. It was evident that cluster 4 displayed notable discrepancies in its preferences for three-stage IFs with different storage periods, with mean scores ranging from 4.70 to 6.91, while cluster 3 had similar preferences for different three-stage IFs.

Comprehensively considering the overall preference and the preference profile of different groups of AHC, it was found that A11 was the least preferred among the one-stage IFs by all consumers, while C11, C10, and C6 received the lowest preferences by approximately 56% of consumers. Conversely, IFs with shorter storage times, including A1, A2, C1, C2, C3, and C5, consistently obtained higher preference scores.

### 3.2. Pivot Profile

For one-stage IFs, the smallest value of all the intensity estimation values was −33 for the aroma term “milky” in the A11 sample. The sum of the absolute value of the lowest frequency (33) was added to the scores of all attributes to generate the contingency table of PP. Results from the CA of the PP contingency table are shown in Figure 2a. The total explained variance for the first two dimensions of CA is 84.92%, with the first and second factors explaining a variability percentage of 77.33% and 7.60%, respectively. The twelve samples’ loadings were spread in the four different quadrants of the symmetric plot (Figure 2a), which indicated they were well discriminated by the consumer using the PP method.

The one-stage samples were most differentiated by the “milky” and “fishy” aroma due to their significant positive and negative correlations along the first dimension (F1 explained 77.33% of the total variance) of the symmetric plot (Figure 2a). The two attributes “milky” and “T-sweet” on the right side (contributions of 50.1% and 15.9%, respectively) and “fishy” on the left side (contribution of 26.5%) contributed greatly to forming the first axis. Therefore, samples located on the right side were mainly positively correlated with “milky” aromas and “T-sweet” tastes, like A2, A1, and A3. A2 had the highest positive quotes for both “milky” aroma (35) and “T-sweet” taste (20). These samples were opposed to the other samples that were characterized as “fishy”, with 66 positive quotes for A11 followed by A9 and A10. In addition, A9, A10, and A11 were positively correlated with the aroma attributes “eggy” and “oxidation” and the flavor term “T-salty”.

In order to explore the relationship between sensory attributes and IF samples for segregating cluster consumers by AHC, CA results were obtained from the cluster 1 and cluster 2 consumers for one-stage IFs (Figure 2b,c), as well as from cluster 3 and cluster 4 for three-stage IFs (Figure 2e,f). As shown in Figure 2b, F1 explained 69.59% of the total variance, with “milky” and “fishy” being the main discriminating attributes. A2 and A3 were characterized by a “milky” aroma, while A11 was positively correlated with a prominent “fishy” aroma. A7 and A14 were associated with a “soy milk” aroma and “T-sweet” taste. In Figure 2c, F1 accounted for 59.18% of the variance, and two attributes, “milky” and “fishy” (contributions of 70.3% and 16.5%, respectively), contributed greatly to form the first axis. A1 was characterized by a “milky” aroma, while A11 was perceived as having a “fishy” aroma. A9 and A10 were associated with the flavor attribute “T-mellow” and the aroma attribute “eggy”.

For three-stage IFs, the smallest value of all the intensity estimation values was −41 for the aroma term “milky” in the C8 sample. A contingency table of the PP test was created by the absolute value (41) plus the intensity estimation values of all terms. Then, the CA plot was obtained and is shown in Figure 2d. It is observed that the first and second dimensions of CA explained 78.43% of the total variance of the experimental data, with the first factor (F1) explaining a variability percentage of 66.36% and the second factor (F2) explaining 12.06%. Additionally, the positions of twelve samples were dispersed in four different quadrants of the CA plot for PP (Figure 2d), which indicated that samples could be distinguished by the PP method using untrained assessors. On the plot, it shows three groups of three-stage Ifs, namely group one, containing C1, C2, C4, C5, and C3, posited near each other between the first and fourth quadrants; group two, containing C6, C11, and C10, spread in the second quadrant; and group three, containing C7, C8, C14, and C9, located in the third quadrant.

As shown in Figure 2d, the aroma terms “milky” and “fishy” of three-stage IFs were most distinguished along the first dimension (contributions of 39.7% and 26.7%, respectively), and were located on the most right and left sides, respectively. C2 and C1 were discovered to be positively correlated with the odor term “milky”, while the opposite samples were characterized by a “fishy” odor with 76 positive quotes for C6 followed by C10 and C11 with 74 and 64 quotes, respectively. C10, C11, and C6 were also positively correlated with the odor term “eggy”. In addition, the flavor term “T-sweet” contributed 11.5% to the first axis, characterizing the samples plotted to the right of the map, including C3, C1, C2, C5, and C4. The attribute “cereal” was more pronounced for C8, as well as a “T-mellow” taste for C3.

Figure 2e,f displays the CA results for the cluster 3 and cluster 4 populations of three-stage IFs, respectively. As shown in Figure 2e, F1 and F2 accounted for 44.69% and 21.25% of the total variance, respectively. The three attributes “milky”, “T-mellow”, and “T-sweet”, located on the right side of the F1 axis, made contribution of 56.0%, 10.8%, and 5.6% for F1, respectively. C1 and C2 were characterized by “milky” aromas, while C4 and C5 were positively correlated with “T-mellow” and “T-sweet” tastes. It is worth noting that, despite the association of C7 and C14 with “fishy” aromas, the proportion of this cluster’s consumers choosing “fishy” was relatively low (Appendix A). In Figure 2f, F1 explained 72.03% of the variance, with “milky”, “fishy” and “T-sweet” (contributions of 35.9%, 34.4%, and 18.5%, respectively) being the main discriminating attributes. C1, C2, and C3 were characterized by “milky” aromas, while C10 and C11 were positively correlated with “fishy” aromas. C5 and C9 were associated with the flavor attribute “T-sweet”.

In general, CA of segregated cluster consumers for one-stage and three-stage IFs indicated that maternal consumers could perceive the variations in sensory attributes “fishy”, “milky”, “T-mellow”, and “T-sweet” among different batches of infant formula, which may have a significant impact on flavor preferences.

### 3.3. Quantitative Descriptive Analysis

The trained panel assessed five aroma attributes and one flavor attribute of various IF samples. The mean scores for each attribute and significant differences among one-stage and three-stage IFs with different storage periods are illustrated in Figure 3a,b. The results obtained from the ANOVA analysis indicated that all six attributes were highly significantly different (*p* < 0.001) among the one-stage IFs (Figure 3a), while “milky”, “creamy”, “fishy”, “oxidation”, and “T-sweet” were found to be highly significantly different (*p* < 0.001) among the three-stage IFs (Figure 3b). However, the panel was unable to differentiate the three-stage IFs through the application of the “eggy” aroma attribute.

In the case of one-stage IFs, the mean scores for the “milky” aroma ranged from 3.86 to 5.09, indicating that the intensity of the “milky” aroma was either slightly weak or moderate (Figure 3a). The highest “milky” aroma was observed in A2, A3, and A6, which exhibited significantly higher scores than the other one-stage IFs. Conversely, the lowest “milky” aroma was found in A4, which had a score below 4. The mean scores for the “creamy” aroma of one-stage IFs exhibited a range of 2.68 to 3.59, indicating a relatively weaker intensity. A3 and A2 had the highest “creamy” aroma, significantly higher than other one-stage IFs. All one-stage IFs displayed a higher intensity of “T-sweet”, with mean scores ranging from 5.95 to 7.23. The highest level of “T-sweet” taste was observed in A2, followed by A5, while A14 was found to exhibit the lowest “T-sweet” taste. The “fishy” aroma exhibited the greatest variation among one-stage IFs with different storage periods, with scores between 1.64 and 4.00. Among all one-stage IFs, A11 achieved the highest score of 4.00 for “fishy” aroma. Additionally, A4, A14, A10, A8, and A9 showed significantly higher scores for “a fishy” aroma than the other one-stage IFs. In contrast, A1, A2, and A3 were observed to have the weakest “fishy” aroma, with scores below 2. All one-stage IFs achieved low scores for the “oxidation” aroma, below 3, with relatively higher scores observed in A4 and A8. This was similar to the results obtained in the consumer-based Pivot Profile (Appendix A). A14 and A10 were found to possess significantly higher “eggy” aromas than the other one-stage IFs. Conversely, A1 exhibited the lowest degree of this aroma, followed by A3.

For three-stage IFs, the scores for the “milky” aroma ranged from 4.05 to 5.64 (Figure 3b), with the highest scores observed in C3 (5.64), followed by C5 (5.36) and C2 (5.18). The “creamy” aroma exhibited minimal variation among the three-stage IFs, with C5, C3, and C2 receiving the highest scores and C6 receiving the lowest rating for “creamy” aroma. The three-stage IFs exhibited a distinct differentiation in “T-sweet” taste, with scores ranging from 5.82 to 7.91. C5 displayed the highest “T-sweet” taste score of 7.91, followed by C1 and C2. In contrast, C14 and C11 were observed to possess a weaker “T-sweet” taste. The most pronounced “fishy” aroma was observed in C6 (4.00), C11 (3.64), and C14 (3.50), while C2 (1.55) and C3 (1.55) exhibited the least perceptible “fishy” aroma. All three-stage IFs presented a weak “oxidation” aroma, with scores ranging from 1.50 to 2.68. C6 was found to achieve the highest score for the “oxidation” aroma, while C2 and C3 received the lowest rating. The ANOVA analysis revealed no statistically significant difference in the “eggy” aroma among the three-stage IFs.

PCA was conducted on the mean values of significantly different attributes; thus, six attributes for one-stage IFs and five attributes, excluding “eggy”, for three-stage IFs were included in the analysis (Figure 4a,b). The first two components collectively explained 87.02% and 94.92% of the variance, respectively, for one-stage and three-stage IFs. PC1 was the major component, accounting for 65.81% and 84.70% of the variance for one-stage and three-stage IFs, respectively. As illustrated by Figure 4a,b, the distribution of sensory attributes for one-stage and three-stage IFs exhibited notable similarity. The aroma attributes “fishy”, “oxidation”, and “eggy” were highly correlated with PC1 in the positive direction. Conversely, the flavor attribute “T-sweet”, as well as the aroma attributes “milky” and “creamy”, correlated with PC1 in the opposing direction. It is worth noting that one-stage or three-stage IFs with shorter storage times, such as A1, A2, and A3, together with C1, C2, C3, and C5, were highly associated with “milky” and “creamy” aromas and “T-sweet” tastes. In contrast, IFs with longer storage times, such as A4, A8, A9, A10, A11, and A14, as well as C6, C10, C11, and C14, exhibited a high correlation with “fishy” and “oxidation” aromas.

### 3.4. External Preferences Analysis

External preference mapping predicted cluster acceptance of one-stage and three-stage IFs, respectively, characterized by descriptive analysis attributes (Figure 5a,b). For one-stage IFs, both cluster 1 and cluster 2 had the highest liking for A1, A2, A3, A6, and A7; each of them was liked by 80% to 100% of both cluster consumers (Figure 5a). These one-stage IFs were characterized by the flavor attribute “T-sweet”, as well as the aroma attributes “creamy” and “milky”. In contrast, A4, A8, A9, A10, A11, and A14 were least preferred by all consumers, and these IFs were associated with “fishy”, “oxidation” and “eggy” aromas.

In the case of three-stage IFs, C1, C2, C3, C4, C5, C7, and C8 were preferred by both cluster 3 and cluster 4, and they were preferred by 80% to 100% of consumers in both clusters (Figure 5b). These three-stage IFs were positively correlated with “creamy” and “milky” aromas, together with “T-sweet” taste. Conversely, all consumers had the least liking for C6, C10, C11, and C14, which they attributed with “fishy”, “oxidation”, and “eggy” aroma attributes.

In general, for either one-stage or three-stage IFs, consumers may prefer IFs with a shorter storage time compared to those with longer storage time. The occurrence of negative aroma attributes, such as “fishy” and “oxidation”, during prolonged storage periods may significantly reduce the consumer preference for IF samples. Additionally, the perception of “milky” and “creamy” aromas and “T-sweet” taste may be critical positive factors influencing consumer preference for IF samples.

## 4. Discussion

This study employed both consumer-based tests and trained evaluation panels to demonstrate the sensory profile variations in one-stage and three-stage IFs with different storage periods, respectively. The results obtained from PP and QDA methods aligned moderately well. The sample space configuration was similar between PP and QDA methods. The IF samples with shorter storage periods were well separated in both methods from those with longer storage periods, which exhibited different aroma and flavor attributes. Indeed, expert evaluators were capable of accurately expressing the sensory attributes of a given product and perceiving differences between products. Nevertheless, even those lacking formal training in evaluation could perceive and depict these distinctions through the data obtained. The findings not only demonstrated the role of untrained evaluators in sensory evaluation but also confirmed the applicability of the Pivot Profile method in analyzing the sensory characteristics of infant formula products, which was consistent with the application of PP in wine [21].

As expected, trained panelists used technical terms to describe some sensory characteristics, such as “oxidation”, while the consumers used more specific equivalents, namely “milky”, “sweet” and “soy milk”. In previous studies, the occurrence of “oxidation” was frequently reported as a result of lipid oxidation in infant formulas with a prolonged shelf life [29,39]. Thus, the “oxidation” aroma attribute was included in the candidate attributes, but its selection ratio was low due to the fact that the term “oxidation” was not easily understood by consumers. It is possible that a trained panel facilitates the more effective characterization of subtle or less prominent attributes, whereas consumers untrained in PP may primarily focus on the most prominent attributes and neglect subtle differences between samples. This similar finding was observed when PP was applied in wine [21].

The “fishy” odor was found to exhibit the greatest variation among different IFs. As the storage period of IF was extended, the “fishy” odor was perceived with greater intensity by both consumers and expert evaluators. Based on correspondence analysis and external preference analysis, it was found that the “fishy” odor was the predominant factor in reducing the preference of consumers for both one-stage and three-stage IFs. This finding was highly consistent with the results of previous studies conducted on infant formula and other foods among the Chinese population [29,40,41]. Previous researchers have conducted extensive investigations into the source of the “fishy” odor in infant formula products. The storage time was found to have a significant impact on the “fishy” odor of IF [6,14]. Although the addition of polyunsaturated fatty acids in formula milk can better meet the nutritional needs of infants and consumers’ expectations, it is susceptible to lipid oxidation, which results in a metallic or fishy odor [14]. This oxidation could occur at any stage, including raw materials [42,43], production and shelf processes [6,14,15,39].

In contrast to the “fishy” odor, “milky” and “creamy” aromas and a “T-sweet” taste were found to be the critical factors influencing consumer preference for IF samples according to the external preference analysis. These findings were in good agreement with our previous studies [36]. Significant variations in “milky” and “T-sweet” were observed among IFs with different storage periods. Considering that the sugar content might not vary significantly in IF, it was speculated that volatile aldehydes and ketones, resulting from the oxidation of unsaturated fatty acids [14,39,44,45], were likely to mask the “milky” odor and reduce the “T-sweet” perception in the mouth. It has been reported that these compounds typically have pronounced odors and their presence might interfere with the perception of sweet taste receptors, leading to a decreased T-sweet sensation [46]. Nevertheless, further research is needed to confirm this hypothesis. Additionally, it should be noted that some IF samples with long storage periods exhibited a pronounced “fishy” odor, but the “milk” odor could also be perceived by consumers. Although the “fishy” odor was found to be negatively correlated with the “milky” odor, it was unable to completely mask the “milky” odor. The balance between these two odors may exert a significant impact on the sensory profile and consumer preferences of IF.

With regard to consumer preference, either one-stage or three-stage IFs with a shorter storage period were more preferred. Consumers usually give preference to fresh products or those close to the manufacturing date, which is also the case for freshly cut watermelon [47], orange juice, and apple juice [48]. According to AHC analysis, two consumer clusters were identified for one-stage and three-stage IFs, respectively, that differed in their liking of IFs. Similar results were observed for dry aged mutton [38], beef [49], and strawberry [50], which consistently showed that consumers displayed differing patterns of preference. Further analysis of the PP data revealed that only 43% of cluster 3 consumers perceived C6, C10, and C11 as having a stronger “fishy” odor compared to the pivot sample, whereas more than 80% consumers in cluster 4 perceived these samples as having a stronger “fishy” odor. It could be speculated that cluster 4 may be more sensitive to sensory changes among IFs with different storage periods than cluster 3, accompanied by a low tolerance for off-flavors, such as “fishy”. This indicated that consumers may exhibit disparate preferences with regard to infant formula. Therefore, it is essential to ensure an adequate sample size of consumers when conducting preference tests, in order to prevent potential biases in the results.

## 5. Conclusions

This study investigated the sensory characteristics and consumer preferences of one-stage and three-stage infant formula products with a storage period spanning from 1 to 14 months using a combination of Pivot Profile (PP), Quantitative Descriptive Analysis (QDA), and consumer preference tests. The PP and QDA results aligned moderately well. The attributes “milky”, “T-mellow”, and “T-sweet” exhibited the greatest variations among IFs with different storage periods. Regardless of one-stage or three-stage, IFs with shorter storage times, such as A1, A2, and A3, together with C1, C2, C3, and C5, were highly associated with “milky” and “creamy” aromas and “T-sweet” taste. In contrast, IFs with longer storage times, such as A4, A8, A9, A10, A11, and A14, as well as C6, C10, C11, and C14, exhibited a high correlation with “fishy” and “oxidation” aromas. The consumer preference tests indicated that IFs with shorter storage times consistently obtained higher preference scores. The results of external preference analysis revealed that IFs with longer storage times were the least preferred by most consumers, which was attributed to the occurrence of some negative aroma attributes, such as “fishy” and “oxidation”. Conversely, “milky” and “creamy” aromas and “T-sweet” taste were identified as the main positive factors influencing consumer preference for IF samples.

## Figures and Tables

**Figure 1 foods-13-02839-f001:**
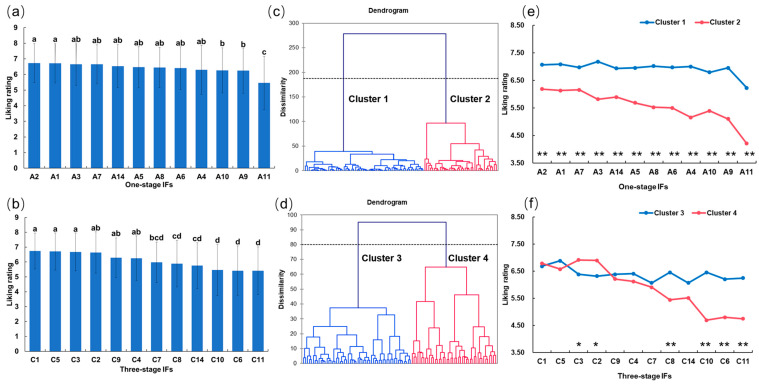
Mean preference ratings for 12 batches of (**a**) one-stage IFs and (**b**) three-stage IFs from 100 consumers. Different letters in the column indicate significant differences among samples at the level of *p* < 0.05. AHC results of preference data for 12 batches of (**c**) one-stage IFs and (**d**) three-stage IFs. Mean preference ratings for 12 batches of (**e**) one-stage IFs and (**f**) three-stage IFs of the two identified consumer clusters. *p* < 0.01 **; *p* < 0.05 *.

**Figure 2 foods-13-02839-f002:**
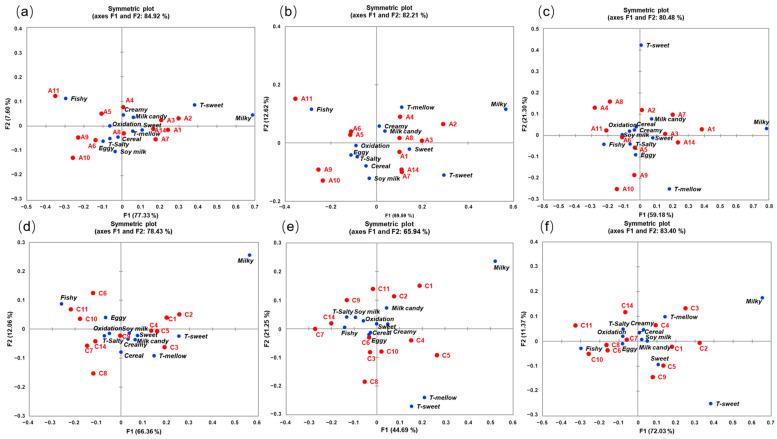
Correspondence analysis biplot of the 12 batches of one-stage IFs using PP from (**a**) all 100 consumers; (**b**) the 62 consumers in cluster 1; (**c**) the 38 consumers in cluster 2. Correspondence analysis biplot of the 12 batches of three-stage IFs using PP from (**d**) all 100 consumers; (**e**) the 44 consumers in cluster 3; (**f**) the 56 consumers in cluster 4. IF samples are shown in red, and attributes are shown in blue. Attributes with “T” represent taste attributes, while those without represent aroma attributes.

**Figure 3 foods-13-02839-f003:**
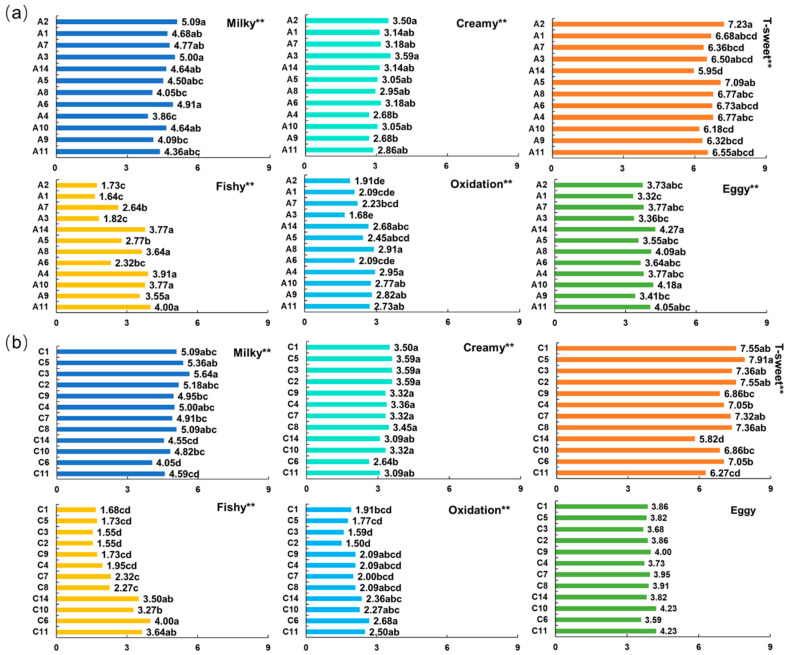
The average intensity of different sensory attributes for 12 batches of (**a**) one-stage IFs and (**b**) three-stage IFs from QDA results. Different letters in the column indicate significant differences among samples (*p* < 0.01 **).

**Figure 4 foods-13-02839-f004:**
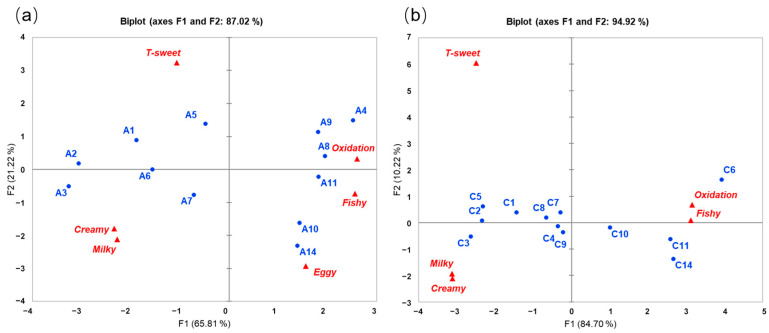
PCA scores and loadings biplot of mean sensory ratings for 12 batches of (**a**) one-stage IFs and (**b**) three-stage IFs from QDA. IF samples are shown in blue, and attributes are shown in red. Attributes with “T” represent taste attributes, while those without represent aroma attributes.

**Figure 5 foods-13-02839-f005:**
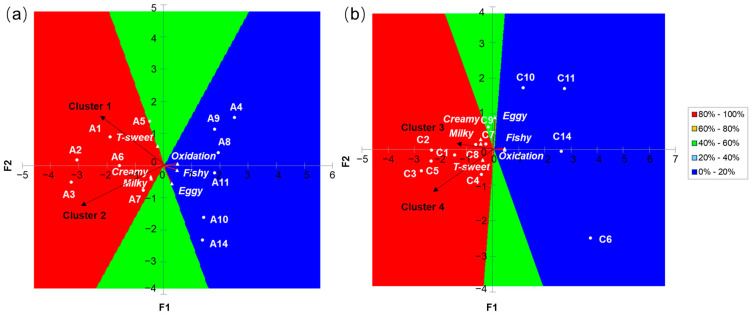
External preference map for 12 batches of (**a**) one-stage IFs and (**b**) three-stage IFs.

**Table 1 foods-13-02839-t001:** The specific information of the IF samples.

NO.	Sample ^a^	IF Stage	Storage Period (Months)	Production Date	Production Specification
1	A1	IF Stage 1	1	4 August 2022	700 g/can
2	A2	IF Stage 1	2	15 July 2022	900 g/can
3	A3	IF Stage 1	3	18 June 2022	708 g/can
4	A4	IF Stage 1	4	17 May 2022	700 g/can
5	A5	IF Stage 1	5	7 April 2022	700 g/can
6	A6	IF Stage 1	6	16 March 2022	700 g/can
7	A7	IF Stage 1	7	6 February 2022	700 g/can
8	A8	IF Stage 1	8	2 January 2022	708 g/can
9	A9	IF Stage 1	9	10 December 2021	700 g/can
10	A10	IF Stage 1	10	21 November 2021	700 g/can
11	A11	IF Stage 1	11	13 October 2021	708 g/can
12	A14	IF Stage 1	14	4 July 2021	700 g/can
13	R-1	IF Stage 1	1	28 July 2022	700 g/can
14	C1	IF Stage 3	1	10 August 2022	700 g/can
15	C2	IF Stage 3	2	2 July 2022	700 g/can
16	C3	IF Stage 3	3	26 June 2022	700 g/can
17	C4	IF Stage 3	4	3 May 2022	700 g/can
18	C5	IF Stage 3	5	17 April 2022	700 g/can
19	C6	IF Stage 3	6	4 March 2022	700 g/can
20	C7	IF Stage 3	7	14 February 2022	700 g/can
21	C8	IF Stage 3	8	29 January 2022	700 g/can
22	C9	IF Stage 3	9	21 December 2021	700 g/can
23	C10	IF Stage 3	10	9 November 2021	700 g/can
24	C11	IF Stage 3	11	20 October 2021	700 g/can
25	C14	IF Stage 3	14	11 July 2021	700 g/can
26	R-3	IF Stage 3	1	14 August 2022	700 g/can

^a^ Sample code A represents one-stage IF, and sample code C represents three-stage IF; the smaller the number, the newer the production batch, and the larger the number, the older the production batch. R-1 and R-3 are pivot samples for one-stage and three-stage IFs, respectively, in the Pivot Profile tests.

**Table 2 foods-13-02839-t002:** The definitions, references, and corresponding intensities of the selected IF sensory descriptors.

Descriptor	Definition	Reference	Intensity ^a^
Milky [33]	Characteristic aroma associated with raw milk	100% Junlebao fresh Delicious Pasteurized Milk	*n* = 8
Creamy [34]	The aroma of mixed cream and pure milk	0.5 mL of Milkground thin cream added to 30 mL of Mengniu skimmed milk	*n* = 5
Fishy [35]	The fishy smell of living fish	20 mL purified water + 0.4 g DHA powder + 0.2 g ARA powder	*n* = 7
Eggy [35]	The aroma of salted egg yolk	0.01 g eggy essence added to 30 mL purified water	*n* = 5
Oxidation [36]	Smell similar to rancidity of oil	30 mL purified water + 0.2 µL 2,4-nonadialdehyde + 0.7 µL 3-methylbutyraldehyde	*n* = 6
T-Sweet [16]	Basic taste sense	64 g/L granulated sugar	*n* = 9

^a^ Defined by the evaluator with reference to the n-butanol and sugar scale, the strength can be corrected based on the n-butanol and sugar scale.

## Data Availability

The original contributions presented in this study are included in the manuscript/Appendix A; further inquiries can be directed to the corresponding author.

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
