# Peer review of "Variations in the Sensory Attributes of Infant Formula among Batches and Their Impact on Maternal Consumer Preferences: A Study Combining Consumer Preferences, Pivot Profile, and Quantitative Descriptive Analysis"

_foods, 2024, doi:10.3390/foods13172839_

Round 1

Reviewer 1 Report

Comments and Suggestions for Authors

The study titled "Variations in Sensory Attributes of Infant Formula among Batches and Their Impact on Maternal Consumer Preferences: A Study Combining Consumer Preferences, Pivot Profile, and Quantitative Descriptive Analysis" contains some methodological limitations that need to be clarified and improved for it to be considered for publication.

Points that need to be clarified in the text:

  • The exact number of samples evaluated in each sensory analysis session (acceptance, QDA, and Pivot Profile).

  • Line 154: "For evaluation of IF products, assessors were instructed to first smell, and then taste IF samples." Why was visual inspection excluded from the test? Please provide justification.

  • Lines 158-161: "The consumer test was evaluated using the 9-point preference scale according to the 'Guidelines for the use of quantitative response scales for sensory analysis' (GB/T 39501-2020). The scores were assigned the following values: 1 = dislike extremely, 2 = dislike very much, 3 = dislike, 4 = dislike somewhat, 5 = dislike not really, 6 = like somewhat, 7 = like, 8 = like very much, 9 = like extremely." This scale is uncommon in sensory evaluations, especially due to its imbalance. Therefore, please add references to robust studies that have used this scale.

  • Additionally, concerning the scale used in the consumer test, reference [35] used to support it is not valid, as the cited study did not use this scale.

  • Was the list of attributes provided to consumers randomized and balanced to avoid bias? [Ares, G., & Jaeger, S. R. (2013). Check-all-that-apply questions: Influence of attribute order on sensory product characterization. Food Quality and Preference, 28(1), 141-153.]

  • Was there any training conducted for the QDA panel?

  • What references were used for the maximum and minimum intensity ratings in the QDA?

  • Were there any replicate sessions in the QDA evaluation?

Author Response

Comments 1: The exact number of samples evaluated in each sensory analysis session (acceptance, QDA, and Pivot Profile).

Response 1: Thanks for your suggestion. We have added the relevant formation in the revised manuscript (Line 178-180, 220-222). Both PP and consumer preference tests were conducted over five days with 20 consumers per day. All IF samples were evaluated in six sessions, with four samples in each session. For QDA, all IF samples were evaluated in duplicate by panelists in twelve sessions of four samples per session over two days.

Comments 2: Line 154: "For evaluation of IF products, assessors were instructed to first smell, and then taste IF samples." Why was visual inspection excluded from the test? Please provide justification.

Response 2: Thanks for your comments. Based on our previous observations, the differences in appearance between different IF samples are minimal. Besides, a large amount of consumer data from the previous period showed that consumer complaints focused on changes in the odor and flavor of infant formula. Previous studies have reported that the aroma compounds of IF samples underwent changes and the corresponding changes in sensory characteristics had an impact on consumer acceptance during shelf-life [1,2]. Thus, our study focused on the odor and flavor of IF samples.

References

1.          Li, Y.; Li, R.; Hu, X.; Liu, J.; Liu, G.; Gao, L.; Zhang, Y.; Wang, H.; Zhu, B. Changes of the volatile compounds and odors in one-stage and three-stage infant formulas during their secondary shelf-life. Curr. Res. Food Sci. 2024, 8, 100693. https://doi:10.1016/j.crfs.2024.100693.

2.          Li, Y.; Wang, H.; Liu, G.; Shi, B.; Zhu, B.; Gao, L.; Zhong, K.; Zhang, Y.; Zhao, L.; Li, R.; et al. An assessment of the sensory drivers influencing consumer preference in infant formula, assessed via sensory evaluation and GC-O-MS. Food Chem. 2024, 455, 139881. https://doi:10.1016/j.foodchem.2024.139881.

Comments 3: Lines 158-161: "The consumer test was evaluated using the 9-point preference scale according to the 'Guidelines for the use of quantitative response scales for sensory analysis' (GB/T 39501-2020). The scores were assigned the following values: 1 = dislike extremely, 2 = dislike very much, 3 = dislike, 4 = dislike somewhat, 5 = dislike not really, 6 = like somewhat, 7 = like, 8 = like very much, 9 = like extremely." This scale is uncommon in sensory evaluations, especially due to its imbalance. Therefore, please add references to robust studies that have used this scale.

Response 3: Thank you for pointing this out. We apologize for the inaccurate translation of 9 points and have made corrections on the meaning of scores: 1 = dislike extremely, 2 = dislike very much, 3 = dislike moderately, 4 = dislike slightly, 5 = neither like nor dislike, 6 = like slightly, 7 = like moderately, 8 = like very much, 9 = like extremely (Line 159-161). The 9-point hedonic scale has been widely used in consumer-based sensory research [3-5]. We have provided the references on this scale in the revised manuscript (Line 159-161).

References:

3.          Wichchukit, S.; O'Mahony, M. The 9-point hedonic and unstructured line hedonic scales: An alternative analysis with more relevant effect sizes for preference. Food Qual. Prefer. 2022, 99, 104575. https://doi:10.1016/j.foodqual.2022.104575.

4.          Xia, Y.; Song, J.; Zhong, F.; Halim, J.; O'Mahony, M. The 9-point hedonic scale: Using R-Index Preference Measurement to compute effect size and eliminate artifactual ties. Food Qual. Prefer. 2020, 133, 109140. https://doi:10.1016/j.foodres.2020.109140.

5.          Kalva, J.J.; Sims, C.A.; Puentes, L.A.; Snyder, D.J.; Bartoshuk, L.M. Comparison of the hedonic general labeled magnitude scale with the hedonic 9‐Point Scale. J. Food Sci. 2014, 79, S238-S245. https://doi:10.1111/1750-3841.12342.

Comments 4: Additionally, concerning the scale used in the consumer test, reference [35] used to support it is not valid, as the cited study did not use this scale.

Response 4: Thank you for pointing this out. We apologize for the incorrect citation of references and have deleted the relevant sentence in the revised manuscript (Line 239).

Comments 5: Was the list of attributes provided to consumers randomized and balanced to avoid bias? [Ares, G., & Jaeger, S. R. (2013). Check-all-that-apply questions: Influence of attribute order on sensory product characterization. Food Quality and Preference, 28(1), 141-153.]

Response 5: Thanks for your suggestion. In PP test, a list of sensory attributes were provided for consumers as a reference, and we encourage consumers to use other sensory attributes freely. The ordering of sensory attributes were not considered in this study. We consider this is a factor of concern and will take it into account in subsequent experiments.

Comments 6: Was there any training conducted for the QDA panel?

Response 6: Thanks for your suggestion. We have added relevant information of QDA evaluation (Line 200-203). All panelists had participated in twelve training sessions (1.5 h each) lasting two months, covering a range of sensory evaluation techniques and methods. These included aroma recognition, triangle tests, ranking, attribute generation, agreement on reference standards, and intensity rating of targeted attributes.

Comments 7: What references were used for the maximum and minimum intensity ratings in the QDA?

Response 7: Thanks for your comments. We have added the relevant information in the revised manuscript (Line 209-211, Supplementary Table S1). The panelists were trained to use a calibrated n-butanol scale for scaling odor intensities [6] and a calibrated granulated sugar scale for flavor intensity ratings. As shown in Table S1, the standard water solutions of n-butanol and granulated sugar were prepared respectively according to the odor and flavor intensity referencing scales. Thus, the intensity ratings of sensory attributes can be corrected with reference to the n-butanol scale and granulated sugar scale.

Table S1 Odor and flavor intensity referencing scales

Intensity level

9-point scale

1

2

3

4

5

6

7

8

9

n-butanol (μg/L)

20

40

80

160

320

640

1280

2560

5120

sugar (g/L)

0.25

0.5

1

2

4

8

16

32

64

References:

6.          Xiao, Z.; Xiang, P.; Zhu, J.; Zhu, Q.; Liu, Y.; Niu, Y. Evaluation of the perceptual interaction among sulfur compounds in mango by Feller’s additive model, odor activity value and Vector model. J. Agricul. Food Chem. 2019, 67(32), 8926-8937. https://doi: 10.1021/acs.jafc.9b03156.

Comments 8: Were there any replicate sessions in the QDA evaluation?

Response 8: Thanks for your suggestion. We have added relevant information of QDA evaluation (Line 220-221): All IF samples were evaluated in duplicate by panelists in twelve sessions of four samples per session over two days.

Reviewer 2 Report

Comments and Suggestions for Authors

The manuscript is scientifically interesting, it contains a new approach to food quality assessment using sensory methods. Sensory characteristics and consumer preferences for milk products for infants aged 0–6 months (IF stage 1) and for toddlers aged 12–36 months (IF stage 3) were examined. The study aimed to demonstrate the characteristics that differentiate both products, which may raise some doubts because, in principle, these products differ due to their adaptation to the age of children. In addition, the effect of storing both products from 1 to 14 months was assessed, using a combination of Pivot Profile (PP), Quantitative Descriptive Analysis (QDA), and consumer preference tests. The use of appropriate methods for developing and discussing the obtained results, including hierarchical clustering to identify consumer clusters with preferences in the same direction, also deserves special recognition.

The manuscript is quite well-prepared and organized. One of the major shortcomings is the lack of information on the origin (production) of the IF samples. Were these samples purchased, supplied by the manufacturer, or prepared in the laboratory? The first subsection of the methodology provides ingredients, but there is no information on how the samples for evaluation were prepared from them. The table provides packages of about 700-900 grams, and 50 g packages were opened for testing. How and when were these samples prepared? Were the commercial samples in cans taken directly from production and used to prepare small packages that were stored for the indicated time?

In addition, quite relevant articles were used to prepare the Introduction and subsequent parts of the manuscript, including the latest ones from recent years. The citations cover a wide range of foods to which similar sensory evaluation methods have been applied. However, I would suggest considering these citations in some places, because sometimes they are a bit out of place with the subject matter, e.g. lines 88-90 in the Introduction: "Through external preference analysis, Bowen et al. (2019) determined that crispness and juiciness are key driving factors affecting consumers’ preferences for apples [27]."

There is some repetition of information regarding the discussion of results and discussions, which is understandable, but it may be worth improving it slightly.

The caption of Table 1 is not sufficient. The designations of milk samples "One-stage" and "Three-stage" in Table 1 and the manuscript text are inappropriate. I would rather suggest "IF Stage 1" and in Table 1, the title of the column "IF Stage", to link the type of samples closely to the age of the children for whom the milk is intended.

Lines 233-234: „A 9-point preference scale was used to determine the overall preference difference of one-stage and three-stage IFs.” – Sentence to be modified or rather unnecessary here, because information about it is in the methodology. What is the purpose of showing the features that differentiate these two products? In addition, there is no information about the composition of the products, so there is no possibility of justifying the differences.

Figures 2 and 4 should be unified. One has lines connecting the red dots, the other does not. I would suggest changing the scale ranges on both axes of all graphs, e.g. from -0.7 to 0.7 (Fig. 2) and from -7 to 7 (Fig. 4).

Comments on the Quality of English Language

The English language sounds quite good, but I suggest minor improvements, especially the commas.

Author Response

Comments 1: The manuscript is quite well-prepared and organized. One of the major shortcomings is the lack of information on the origin (production) of the IF samples. Were these samples purchased, supplied by the manufacturer, or prepared in the laboratory? The first subsection of the methodology provides ingredients, but there is no information on how the samples for evaluation were prepared from them. The table provides packages of about 700-900 grams, and 50 g packages were opened for testing. How and when were these samples prepared? Were the commercial samples in cans taken directly from production and used to prepare small packages that were stored for the indicated time?

Response 1: Thank you for pointing this out. We have added the information of the origin of the IF samples in the revised manuscript (Line 109-110). All IF samples were obtained from local suppliers as commercial products. The samples with different bathes had different production dates and storage periods as shown in Table 1. All the commercial samples were opened on the same day, and then 50 g of each sample was sampled for testing.

Comments 2: In addition, quite relevant articles were used to prepare the Introduction and subsequent parts of the manuscript, including the latest ones from recent years. The citations cover a wide range of foods to which similar sensory evaluation methods have been applied. However, I would suggest considering these citations in some places, because sometimes they are a bit out of place with the subject matter, e.g. lines 88-90 in the Introduction: "Through external preference analysis, Bowen et al. (2019) determined that crispness and juiciness are key driving factors affecting consumers’ preferences for apples [27]."

Response 2: Thanks for your suggestion. We have deleted the citation "Through external preference analysis, Bowen et al. (2019) determined that crispness and juiciness are key driving factors affecting consumers’preferences for apples [27]." in the revised manuscript.

Comments 3: There is some repetition of information regarding the discussion of results and discussions, which is understandable, but it may be worth improving it slightly.

Response 3: Thanks for your suggestion. We have made improvement on results and discussion in the revised manuscript.

Comments 4: The caption of Table 1 is not sufficient. The designations of milk samples "One-stage" and "Three-stage" in Table 1 and the manuscript text are inappropriate. I would rather suggest "IF Stage 1" and in Table 1, the title of the column "IF Stage", to link the type of samples closely to the age of the children for whom the milk is intended.

Response 4: Thanks for your suggestion. We have changed the title of column "Stage", "One-stage" and "Three-stage" to the title of column "IF Stage", “IF Stage 1” and “IF Stage 3” in Table 1.

Comments 5: Lines 233-234: „A 9-point preference scale was used to determine the overall preference difference of one-stage and three-stage IFs.” – Sentence to be modified or rather unnecessary here, because information about it is in the methodology. What is the purpose of showing the features that differentiate these two products? In addition, there is no information about the composition of the products, so there is no possibility of justifying the differences.

Response 5: Thanks for your suggestion. We have deleted the sentence “A 9-point preference scale was used to determine the overall preference difference of one-stage and three-stage IFs”. The expression of this sentence is not quite appropriate. A 9-point preference scale was used to determine the overall preference of different batches of one-stage and three-stage IF, respectively. The main purpose of this study was to explore the different sensory features among batches instead of two products.

Comments 6: Figures 2 and 4 should be unified. One has lines connecting the red dots, the other does not. I would suggest changing the scale ranges on both axes of all graphs, e.g. from -0.7 to 0.7 (Fig. 2) and from -7 to 7 (Fig. 4).

Response 6: Thanks for your suggestion. We have removed the lines in Figure 4 in the revised manuscript. However, we don't think it's the best presentation to scale all graphs to a uniform scale. So, we have kept the current graphs to better present the results.

4. Response to Comments on the Quality of English Language

Point 1: The English language sounds quite good, but I suggest minor improvements, especially the commas.

Response 1: Thank you for your comments. The revised manuscript was edited to ensure that the language is clear and free of errors by a native English speaker.
